# Improving Prostatic Preoperative Volume Estimation and Planning before Laser Enucleation

**DOI:** 10.3390/jpm12111761

**Published:** 2022-10-25

**Authors:** Ziv Savin, Snir Dekalo, Haim Herzberg, Reuben Ben-David, Yuval Bar-Yosef, Avi Beri, Ofer Yossepowitch, Mario Sofer

**Affiliations:** 1The Department of Urology, Tel Aviv Sourasky Medical Center, Sackler School of Medicine, Tel Aviv University, 6 Weizmann Street, Tel Aviv 6423906, Israel; 2The Endourology Unit, Tel Aviv Sourasky Medical Center, Sackler School of Medicine, Tel Aviv University, Tel Aviv 6423906, Israel

**Keywords:** laser enucleation, prostate volume, ultrasound

## Abstract

We aimed to validate a formula for improving the estimation of prostatic volume by abdominal ultrasound (AUS) prior to transurethral laser enucleation. A total of 293 patients treated for benign prostate hyperplasia (BPH) by laser enucleation from 2019–2022 were included. The preoperative AUS volume was adjusted by the formula 1.082 × Age + 0.523 × AUS − 53.845, which was based on specimens retrieved by suprapubic prostatectomy. The results were compared to the weight of the tissue removed by laser enucleation as determined by the intraclass correlation coefficient test (ICC). The potential impact of preoperative planning on operating time was calculated. The ICC between the adjusted volumes and the enucleated tissue weights was 0.86 (*p <* 0.001). The adjusted volume was more accurate than the AUS volume (weight-to-volume ratio of 0.84 vs. 0.7, *p <* 0.001) and even more precise for prostates weighing >80 g. The median operating time was 90 min. The adjusted volume estimation resulted in an overall shorter expected preoperative operating time by a median of 21 min (24%) and by a median of 40 min in prostates weighing >80 g. The adjustment formula accurately predicts prostate volume before laser enucleation procedures and may significantly improve preoperative planning, the matching of a surgeon’s level of expertise, and the management of patients’ expectations.

## 1. Introduction

Ineffective medical therapy and secondary complications are the leading indications for surgical treatment of benign prostatic hyperplasia (BPH). Transurethral and minimally invasive procedures are indicated for a prostate with a volume of less than 80 cc, while open prostatectomy is reserved mostly for higher-volume prostates [1,2]. With the introduction of the holmium laser in the treatment of BPH, the last 25 years have witnessed continuous progress, expansion, and interest in the performance of transurethral enucleating procedures for BPH, including holmium laser enucleation of the prostate (HoLEP) and thulium laser enucleation of the prostate (ThuLEP), which are both considered size-independent [2,3]. The pre-intervention establishment of BPH volumes is necessary for choosing the appropriate type of surgery, as well as for logistic planning, including instrumentation, anesthesia, and time in the operating room. Estimation of the preoperative BPH volume, however, is not always reliable and commonly relies on digital rectal examination (DRE) and sonography-based imaging studies, which carry a high level of subjectivity and inter-examiner variability [4]. While estimations by transrectal ultrasonography (TRUS) are more accurate than those by abdominal ultrasonography (AUS), AUS is much more widespread and performed more frequently in daily practice [5]. The discrepancy between AUS volume estimation and intraoperative findings, however, can sometimes be very pronounced.

Based upon pathologic specimens during an open prostatectomy, an AUS prostate volume-adjusting formula was calculated to improve the accuracy of AUS volume determinations [6]. Considering the global trend toward shifting from open enucleation to transurethral enucleation, this is a validation study to investigate the accuracy of that formula in predicting prostate adenoma volumes and in improving preoperative planning before transurethral laser enucleations of BPH.

## 2. Methods

### 2.1. Study Population

After receiving the institutional review board’s approval for the *Tel Aviv Sourasky Medical Center* (TLV0610-21, 12 October 2022), we retrospectively reviewed the medical records of 293 consecutive patients who underwent HoLEP and ThuLEP in our institution between 1 January 2020 and 31 July 2022. After excluding patients with incomplete data or insufficient follow-up (*n* = 8) and those converted to suprapubic prostatectomy (*n* = 3), the final study cohort consisted of 282 patients. The demographic, clinical, surgical, and pathological characteristics were recorded, including age, AUS prostatic volume (cc), operating time (minutes), weight of enucleated tissue, as reported on the pathological reports (grams), hospital stay (days), and 30-day postoperative complications, according to the Clavien–Dindo (CD) classification [7].

### 2.2. Volume Estimation

The AUS whole gland prostate volume was estimated and calculated by radiologists using the ellipsoid volume formula (length × width × height × π/6) [8]. The conversion of volume to weight was performed with the conversion of 1 cc (AUS) = 1 g [9,10]. The adjusted volume estimations were performed by means of the following formula: 1.082 × Age + 0.523 × AUS − 53.845 [6].

### 2.3. Surgical Technique

All of the study patients underwent transurethral laser enucleation by a single surgeon following an en-bloc early external sphincter release technique, with the patient under either general or spinal anesthesia [11]. The laser setting for HoLEP was 2 j/50 Hz/short-pulse enucleation and 1.2 j/30 Hz/long-pulse coagulation, and 90 w enucleation and 60 w coagulation for ThuLEP. The procedure includes the following: a direct-vision transurethral insertion of a 26 FR laser resectoscope; a formal cystoscopy to exclude bladder tumors and to identify ureteral orifices; a circular urethral mucosal incision proximal to the verumontanum posteriorly and the formation of a small mucosal flap proximal to the external sphincteric mucosal fold anteriorly; the development of an enucleation plane between the adenoma and the surgical capsule with early upside-down rotation of the resectoscope; an en-bloc enucleation and pushing of the entire adenoma into the bladder while controlling bleeding with a laser; a morcellation and evacuation of the enucleated prostate; the placement of a 20–22 Fr triple-lumen Foley catheter with a balloon inflated in the bladder under mild traction with continuous irrigation. When necessary, additional mono- or bi-polar fulgurations were performed before the catheterization in order to ensure control of hemostasis. The length of the postoperative hospital stays depended upon the clarity of irrigation and the patient’s performance status. In cases with clear irrigation and good performance, the catheter was removed, and after spontaneous voiding, the patients were discharged on the same day. Otherwise, home discharge was postponed until the following morning.

### 2.4. Statistical Analysis

Descriptive statistics were used to summarize the patients’ characteristics. The continuous variables were reported as a median and interquartile range (IQR) and compared by means of the Wilcoxon signed-rank or the Mann–Whitney tests. The categorical variables were reported as numbers (%). The intraclass correlation coefficient was calculated to validate the accuracy of the adjustment formula volume estimation compared to the enucleated tissue weights. The ratios of the enucleated tissue weight-to-volume estimations were used to compare the accuracy of the estimation methods, with a ratio of 1 being considered a perfect match. A subanalysis was performed for prostate weights of >80 g. All of the statistical analyses were two-sided, and significance was defined as *p* < 0.05. The SPSS software (IBM SPSS Statistics, Version 25, IBM Corp., Armonk, NY, USA) was used for all of the statistical analyses.

## 3. Results

The patients’ demographics and clinical characteristics are summarized in Table 1. The median age of the cohort was 73 years (range 48–101), and the median AUS-estimated whole gland prostatic volume was 80 cc (range 20–254). Forty-three patients (15%) underwent additional surgery during the same operative session, including cystolithotripsy and ureteroscopy. The median operating time for the prostatic intervention alone was 90 min (range 30–250), and the median hospital stay for those patients was 1 day (range 1–10). Most of the complications were minor (CD ≤ 2). Eight patients (3%) sustained major postoperative complications (CD ≥ 3) and all of them were treated by endoscopic intervention, except for one who was admitted to the intensive care unit due to pulmonary edema (CD = 4). The median weight of the enucleated tissue was 60 g (range 10–185).

The median adjusted volume, as estimated by the formula, was 67 mL (range 11–157). The intraclass correlation coefficient (ICC) between the adjusted volume and the enucleated tissue weight was 0.86 (95% confidence interval 0.83–0.89, *p* < 0.001). The comparison of adjusted volumes to enucleated tissue weights demonstrated a median weight-to-volume ratio of 0.84 (IQR 0.68–1), which was significantly more accurate than the AUS volume estimations (0.84 vs. 0.7, respectively, *p* < 0.001, Figure 1a). The adjusted volume estimation was more accurate for the prostate weights of >80 g (*N* = 86) compared to the smaller prostates (*N* = 196), with median weight-to-volume ratios of 1.14 (IQR 0.96–1.3) and 0.77 (IQR 0.6–0.92), respectively (*p* < 0.001, Figure 1b).

The median weight of the tissues that had been removed during surgery in 1 min was 0.63 g/min (IQR 0.44–0.86). Based upon this result, the overall paired model demonstrated that the median time difference between the AUS and the formula-adjusted expected operative times was 21 min (IQR 6–45), meaning 24% less. The application of this model to prostates of >80 g revealed that the time difference was significantly greater compared to that for smaller prostates, reaching a median of 40 min (IQR 17–57, *p* < 0.001) (Figure 2).

## 4. Discussion

There is a robust body of literature supporting laser enucleation as surpassing other surgical BPH procedures [3]. Long-term subjective and objective outcomes, including detrusor pressure at maximum flow, erectile function, weight of the removed prostate, and reoperation rates after laser enucleations, have been shown to be superior to transurethral resections [12,13,14]. In addition, there are data indicating that laser enucleation and open prostatectomy are equally efficient and yield similar functional improvements, although those data highlight the advantages of laser transurethral enucleation in terms of blood loss and hospital stay [15,16]. The additional advantages of laser enucleation of BPH, such as safety, efficacy, durability, and less invasiveness, have resulted in growing interest and popularization. Although this approach is size-independent, the prostatic volume is one of the most influential factors in determining the duration of surgery. Recent studies that investigated laser enucleations and assessed the length of the procedure reported a duration of 60–150 min [17,18,19], which is consistent with our results (IQR 75–120 min). The length of operation is also reportedly dependent upon the enucleation technique (en-bloc vs. multi-incisional approaches) and on the surgeon’s skills based upon a learning curve, but mostly it depends upon prostate volume [19,20,21,22]. As such, a reliable preoperative estimation of prostate volume is essential for preoperative planning in terms of operating room logistics, surgeon skills and expertise, time and type of anesthesia, probability of complications, duration of an indwelling catheter, length of hospitalization, and management of patient expectations [19,23,24].

The prostate volume can be estimated by either DRE or imaging studies. While DRE is simple and routinely performed, its correlation with true prostate volumes is poor [4]. Studies based upon the weight of radical prostatectomy specimens revealed that magnetic resonance imaging and computerized tomography are the most precise imaging modalities for assessing prostate volumes [5,25]. However, these imaging studies are indicated for cases of suspected prostate cancer or its staging. Their costs, time consumption, relatively limited availability, and performance by radiologists in dedicated imaging services make them unsuitable for routine use for estimating BPH volume. In addition, these specimens are usually weighed together with the seminal vesicles and the peripheral zone and yield potentially spurious results. In contrast, US is more practical for assessing BPH since it has high availability to both radiologists and urologists, lacks potential risks related to radiation and contrast material, and is significantly less expensive than the other imaging methodologies. TRUS is reportedly more accurate than AUS in predicting prostate volumes and can provide valuable information on prostatic adenomas; however, it is more invasive, inconvenient for the patient, and may be less available [5,25]. It is, therefore, understandable why AUS remains a common tool used for volumetric measurements of BPH before surgery.

Despite its utility being supported by international guidelines, the reliability of AUS in predicting an accurate adenoma size is modest (~65%) [6]. Improving the sonographic estimation of the volume of a prostatic adenoma before an open prostatectomy was suggested by a novel US-based formula, which increased the accuracy by mathematical adjustments using AUS volumes, patient age, and correcting coefficients [6]. The adjusted calculated volumes were more accurate by 20% compared to the enucleated tissue weight. That formula was proposed to optimize the indication of the type of approach (i.e., either transurethral or open). Our rationale for validating and extending its use in the transurethral laser enucleation-oriented environment was based upon the fact that this minimally invasive technique mimics the manual enucleation of prostatic adenomas performed in open prostatectomy. We believe that this concept was satisfactorily validated in the current study by the demonstration of a significant improvement in the prostatic volume estimations and an excellent correlation between the adjusted volumes and the enucleated tissue weights (ICC = 0.86, *p* < 0.001). It is possible that laser enucleation also possesses some ablative effects, resulting in a reduction of the specimen weight, and without this, the correlation could be even more accurate. Since our formula was suitable for both open surgeries and laser enucleations, our results support the idea that laser enucleation is as good as open surgery for removing an entire adenoma. We also believe that using this formula may optimize the preparation for HoLEP and ThuLEP, the use of operating room resources by providing an accurate estimation of surgery duration timings, and improve operative scheduling, as well as the appropriate designation of the surgeon. The formula has been incorporated as a built-in feature on our departmental website (referenced in the Appendix A).

Our study is limited by its retrospective design. In addition, the lack of standardization of bladder status, type of equipment, and interpretation of imaging findings could expose the study to inter- and intra-observer variability bias, as reported in the literature [26]. It also bears mentioning that we relied upon the ellipsoid volume method, while some authors support the use of the “bullet” formula [27]. Finally, we did not separate between the HoLEP and ThuLEP procedures, although the loss of water in the tissues is slightly different between them. However, we believe that the large sample size and the consecutiveness of the cohort mitigate most of these limitations, and that our results may contribute valuable information for the application of the formula in real-life practice.

## 5. Conclusions

The AUS adjustment formula (1.082 × Age + 0.523 × AUS − 53.845) offers an improved and reliable preoperative estimation of prostate volume. Its validation for transurethral enucleation procedures contributes to better logistic preparation and planning and patient expectancy management.

## Figures and Tables

**Figure 1 jpm-12-01761-f001:**
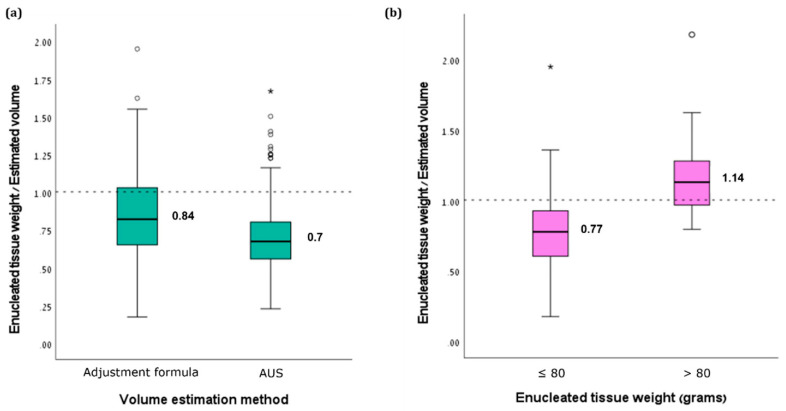
Box-and-whisker plot showing the ratios of enucleated tissue weights and estimated volumes by (**a**) AUS and adjustment formula and by (**b**) adjustment formula among patients with tissue weights above and below 80 g. The dashed horizontal line at level 1.0 represents 100% accuracy. * Represents outliners.

**Figure 2 jpm-12-01761-f002:**
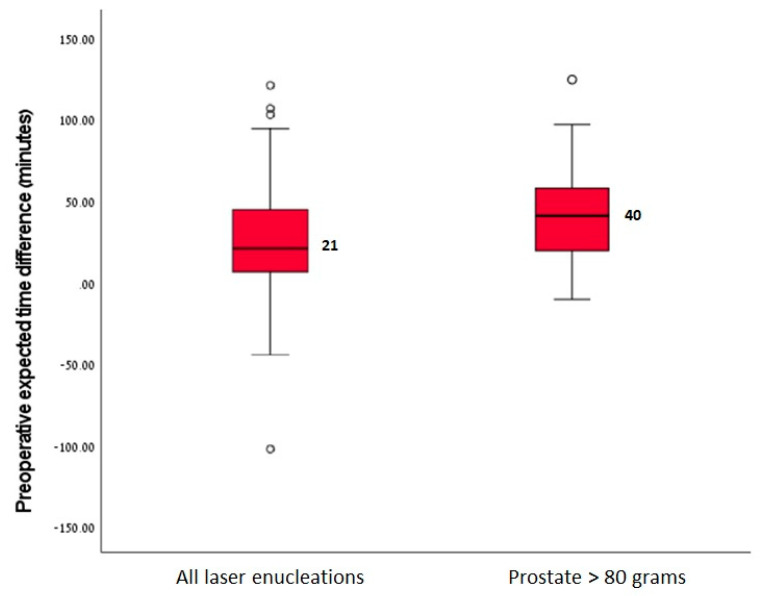
Box-and-whisker plot showing the difference in the expected operative duration between sonographic and adjusted prostate volume estimations.

**Table 1 jpm-12-01761-t001:** Study population characteristics (*n* = 282).

Variable	Value
Age, years; median (range)	73 (68–77)
AUS volume estimation, cc; median (range)	80 (20–254)
Previous BPH surgery, *N*	7 (2%)
Additional same-session operations, *N*	43 (15%)
Cystolithotripsy	23 (8%)
Ureteroscopic lithotripsy	10 (4%)
Transurethral resection of bladder tumor	6 (2%)
Hydrocelectomy	2 (1%)
Percutaneous nephrolithotomy	1 (0.5%)
Open bladder diverticulectomy	1 (0.5%)
Operating time for prostate, minutes; median (range)	90 (30–250)
Hospital stay, days; median (range)	1 (1–10)
Overall postoperative complications, *N*	50 (17%)
Urinary tract infection	9 (3%)
Temporary urinary retention	21 (7%)
Postoperative bleeding needing blood transfusion	5 (2%)
Cardiorespiratory	3 (1%)
Bulbar urethral stricture	3 (1%)
Enucleated tissue weight, grams; median (range)	60 (10–185)

AUS: abdominal ultrasound; BPH: benign prostatic hyperplasia; IQR: interquartile range.

## Data Availability

The data that support the findings of this study are available on request from the corresponding author, [Ziv Savin]. The data are not publicly available due to ethical issues and privacy of the participants.

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
