# Peer review of "Improving Prostatic Preoperative Volume Estimation and Planning before Laser Enucleation"

_jpm, 2022, doi:10.3390/jpm12111761_

Round 1

Reviewer 1 Report

Before transurethral laser enucleation, preoperative transrectal ultrasonography(TRUS) is an essential.

This is not just for measuring the exact volume of the prostate, TRUS can provide valuable information about the prostatic adenoma. This information is an important parameter for surgeons when first learning the procedure. The surgeon should reconstitute the tridimensional configuration of the adenoma by using the transverse and sagittal images of ultrasonography.

Therefore, volume measurement using abdominal ultrasound before transurethral laser enucleation is somewhat unnecessary because transrectal ultrasound is performed. Additionally, I disagree with the authors' statement that abdominal ultrasonography is the most commonly used volume measurement method before prostate surgery.

Although there are no major flaws in the results and conclusions of this study, the goals they pursue in this study do not provide useful information to surgeons.

Author Response

Comment 1:

Before transurethral laser enucleation, preoperative transrectal ultrasonography (TRUS) is an essential. This is not just for measuring the exact volume of the prostate, TRUS can provide valuable information about the prostatic adenoma. This information is an important parameter for surgeons when first learning the procedure. The surgeon should reconstitute the tridimensional configuration of the adenoma by using the transverse and sagittal images of ultrasonography. Therefore, volume measurement using abdominal ultrasound before transurethral laser enucleation is somewhat unnecessary because transrectal ultrasound is performed.

Response 1:

We agree with the reviewer that TRUS is a superior tool for prostate measurements, including measurement of the adenoma, and we mentioned it in the discussion. However, as a preoperative tool, TRUS is invasive and less available in many areas around the world. Thus, abdominal US is a more available as an ambulatory tool for preoperative preparations. In addition, according to the AUA guidelines, prostate size can be assessed using TRUS, abdominal US, cystoscopy, or cross-sectional imaging, prior to intervention, without indicating priorities (Lerner LB, et al. Management of Lower Urinary Tract Symptoms Attributed to Benign Prostatic Hyperplasia: AUA GUIDELINE PART II-Surgical Evaluation and Treatment. J Urol. 2021 Oct;206(4):818-826)

Adenoma measurements by abdominal US is more difficult and radiologists tend to report the overall whole gland prostate volume. Therefore, we suggest that volume adjustment for the abdominal US measurement may contribute for surgeons' preoperative planning.

Changes made ("Discussion" paragraph 2): "TRUS is reportedly more accurate than AUS in predicting prostate volume and can provide valuable information about the prostatic adenoma, however, it is more invasive, inconvenient for the patient and may be less available.5,25 It is therefore understandable why AUS remains a common tool used for volumetric measurements of BPH before surgery. “

Comment 2:

Additionally, I disagree with the authors' statement that abdominal ultrasonography is the most commonly used volume measurement method before prostate surgery.

Response 2: 

See response 1.

Comment 3:

Although there are no major flaws in the results and conclusions of this study, the goals they pursue in this study do not provide useful information to surgeons.

Response 3:

Since abdominal US is still very common as a preoperative tool before BPH surgeries in many areas due to its availability, we think that our goal provides some useful information for surgeons, including: time planning and logistics, surgeon’s expertise, patients’ expectations.  

Reviewer 2 Report

The study deals about the validation of a formula to assess prostate volume.  The weight of resected tissue was compared to the estimated volume by the novel and the standard formula by AUS. The ratio enucleated tissue weight/estimated volume was significantly higher for the novel formula.

1) Authors should disclose that this a validation study

2) It is not clear if the estimated volume refers to the whole gland or to the transitional/central zone of the prostate. As a matter of facts, enucleation is limited to the adenoma and therefore the resected tissue should always be less than the total

Author Response

Comment 1:

Authors should disclose that this a validation study

Response 1: 

Thank you for the comment. We added the information at the "Introduction" section and at a "Disclosure section" at the end.

Changes made ("Introduction" paragraph 2): "this is a validation study to investigate the accuracy of that formula in predicting prostate adenoma volume and in improving preoperative planning before transurethral laser enucleations of BPH."  

Conflicts of Interest Disclosure (at the end): This is a validation study for our developed adjustment formula calculating the estimated prostate adenoma volume.

Comment 2:

It is not clear if the estimated volume refers to the whole gland or to the transitional/central zone of the prostate. As a matter of facts, enucleation is limited to the adenoma and therefore the resected tissue should always be less than the total.

Response 2:

The estimated abdominal US volume referred to the whole gland, and the formula adjusted it for the enucleated adenoma. We made it more clear at the "methods" and "results" sections.

Changes made ("Methods" paragraph 2): "AUS whole gland prostate volume was estimated and calculated by radiologists using the ellipsoid volume formula (length × width × height × π/6).8"

Changes made ("Results" paragraph 1): ."..and the median AUS-estimated whole gland prostatic volume was 80 cc (range 20-254)."

Reviewer 3 Report

This is an interesting study comparing the volume of the prostate by the conventional AUS and an adjusted AUS formula.

Some issues should be addressed:

I could not find the absolute values of the weight of the enucleated tissue. They should be reported.

What is the influence of the so called surgical capsule which is measured by AUS but leaves after enucleation?

As the loss water of the tissue is different between Holium and Thulium laser, the HoLEP and ThuLEP patients should be analyzed separately.

Author Response

Comment 1:

I could not find the absolute values of the weight of the enucleated tissue. They should be reported.

Response 1:

The absolute values of the enucleated tissue are reported and the last sentence of paragraph 1 in the "results" section :"The median weight of the enucleated tissue was 60 grams (range 10-185)." They are also presented in Table 1, last variable. 

Comment 2:

What is the influence of the so called surgical capsule which is measured by AUS but leaves after enucleation?

Response 2:

The estimated abdominal US volume referred to the whole gland (including TZ, CZ, PZ and capsule), and the formula adjusted it for the enucleated adenoma, sparing the surgical capsule which is not enucleated. We made it more clear at the "methods" and "results" sections.

Changes made ("Methods" paragraph 2): "AUS whole gland prostate volume was estimated and calculated by radiologists using the ellipsoid volume formula (length × width × height × π/6).8"

Changes made ("Results" paragraph 1): ."..and the median AUS-estimated whole gland prostatic volume was 80 cc (range 20-254)."

Comment 3:

As the loss water of the tissue is different between Holium and Thulium laser, the HoLEP and ThuLEP patients should be analyzed separately.

Response 3:

Thank you for the contributing comment. We added this in our limitations.

Changes made ("Discussion" paragraph 4): Finally, we did not separate between HoLEP and ThuLEP procedures, although the loss water of the tissue is slightly different between them. 

Round 2

Reviewer 2 Report

None

Reviewer 3 Report

The paper could be accepted now.